# Obesity-Related Genes Expression in Testes and Sperm Parameters Respond to GLP-1 and Caloric Restriction

**DOI:** 10.3390/biomedicines10102609

**Published:** 2022-10-17

**Authors:** Ana S. Correia, Sara C. Pereira, Tiago Morais, Ana D. Martins, Mariana P. Monteiro, Marco G. Alves, Pedro F. Oliveira

**Affiliations:** 1LAQV-REQUIMTE and Department of Chemistry, Campus Universitario de Santiago, University of Aveiro, 3810-193 Aveiro, Portugal; 2Endocrine and Metabolic Research, Unit for Multidisciplinary Research in Biomedicine, Department of Anatomy, School of Medicine and Biomedical Sciences (ICBAS), University of Porto, 4050-313 Porto, Portugal; 3Laboratory for Integrative and Translational Research in Population Health (ITR), University of Porto, 4099-002 Porto, Portugal; 4Department of Pathology, Faculty of Medicine, University of Porto, 4200-319 Porto, Portugal; 5Biotechnology of Animal and Human Reproduction (TechnoSperm), Institute of Food and Agricultural Technology, University of Girona, ES-17003 Girona, Spain; 6Unit of Cell Biology, Department of Biology, Faculty of Sciences, University of Girona, ES-17003 Girona, Spain

**Keywords:** fat mass and obesity, melanocortin-4 receptor, glucosamine-6-phosphate deaminase 2, transmembrane protein 18, caloric restriction, glucagon-like protein 1

## Abstract

**Aim**: Calorie restriction (CR) diets and glucagon-Like Peptide-1 (GLP-1) analogs are known to alter energy homeostasis with the potential to affect the expression of obesity-related genes (ORGs). We hypothesized that CR and GLP-1 administration can alter ORGs expression in spermatozoa and testes, as well as the sperm parameters implicated in male fertility. **Materials and Methods:** Six-week-old adult male Wistar rats (n = 16) were divided into three groups, submitted either to CR (n = 6, fed with 30% less chow diet than the control rats), GLP-1 administration (n = 5, 3.5 pmol/min/kg intraperitoneal) for 28 days, or used as controls (n = 5, fed ad libitum). Selected ORGs expression, namely the fat mass and obesity-associated (*FTO*), melanocortin-4 receptor (*MC4R*), glucosamine-6-phosphate deaminase 2 (*GNPDA2*), and transmembrane protein 18 (*TMEM18*) were evaluated in testes and spermatozoa by a quantitative polymerase chain reaction (qPCR). **Results:** CR resulted in lower body weight gain and insulin resistance, but a higher percentage of sperm head defects. GLP-1 administration, despite showing no influence on body weight or glucose homeostasis, resulted in a lower percentage of sperm head defects. CR and GLP-1 administration were associated with a higher expression of all ORGs in the testes. Under CR conditions, the genes *FTO* and *TMEM18* expression in the testes and the *MC4R* and *TMEM18* transcripts abundance in sperm were positively correlated with the spermatozoa oxidative status. The abundance of *FTO* and *TMEM18* in the spermatozoa of rats under CR were positively correlated with sperm concentration, while the testes’ *TMEM18* expression was also positively correlated with sperm vitality and negatively correlated with insulin resistance. Testes *GNPDA2* expression was negatively correlated with sperm head defects. **Conclusions:** CR and GLP-1 administration results in higher ORGs expression in testes, and these were correlated with several alterations in sperm fertility parameters.

## 1. Introduction

Energy balance is controlled through the hypothalamus’ integration of long-term signals related to energy storage at adipose tissues and meal-related short-term signals. Long-term regulation is associated with the signaling of adiposity hormones, i.e., positively correlated with the amount of body fat [1]. A fine and strict hormonal balance is responsible for energy homeostasis regulation. When this signaling pathway fails, energy balance is dysregulated [1,2,3]. Two peripheral hormones, leptin and insulin, act synergistically with gastrointestinal hormones in the regulation of short-term food intake [4]. Under leptin stimuli, opiomelanocortin neurons express cocaine and amphetamine-related transcript (CART) and α- melanocyte-stimulating hormone (α-MSH), promoting satiety [5]. Meanwhile, insulin, the anabolic hormone, promotes an increase in the rate of glucose transport and glycolytic machinery, ultimately promoting the synthesis of carbohydrates, fat, and proteins [6]. Gastrointestinal hormones are also important regulators of energy balance. Ghrelin is a gastrointestinal hormone that has an orexigenic effect and is considered a short-term signal of energy insufficiency [7]. Ghrelin promotes feeding by Neuropeptide Y (NPY)/ Agouti-related protein (AgRP) neuron activation, leading to increased food intake, body weight, and adiposity [7]. Upon food ingestion, ghrelin secretion is suppressed and other hormones are released from the intestine, including glucagon-like peptide (GLP-1), exerting anorexigenic effects within the hypothalamus [7]. GLP-1 is a peptide that acts as a determinant key in blood glucose homeostasis by its actions in slowing gastric emptying, enhancing pancreatic insulin secretion, and suppressing pancreatic glucagon secretion [8]. Thus, based on GLP-1 proprieties and physiological effects, longer-acting GLP-1 analogs have been developed to be used as pharmacological tools for type 2 diabetes (T2D) and obesity treatment [9]. In fact, the glucose-lowering and weight-loss efficacy of GLP-1 analogs, such as liraglutide and semaglutide, are well demonstrated, besides improving glucose tolerance and decreasing insulin resistance [10,11,12]. Additionally, treatment with liraglutide for weight loss and maintenance was also shown to impact male reproductive potential by increasing sperm counts [13]. Furthermore, GLP-1 was demonstrated to interact with the male reproductive function by decreasing the glucose uptake by Sertoli cells, the somatic cells that provide the nutritional support of spermatogenesis [14].

Caloric restriction (CR) diets are recommended to achieve weight loss and glycemic control in patients with obesity or T2D with concomitant overweight [15,16]. Nevertheless, the impact of CR on the body’s energy-demanding functions, such as male reproductive function, remains largely unknown. Energy homeostasis disorders, including severe malnutrition or obesity, can be associated with male reproductive dysfunction [17]. Indeed, both a low and high body mass index (BMI), under 19 kg/m^2^ or over 30 kg/m^2^, have been associated with reduced testicular volume and semen quality, suggesting impaired spermatogenesis [18]. Additionally, a low BMI is associated with altered semen parameters, including a decreased total sperm count and semen volume, rather than sperm concentration and overall or progressive motility [19]. Thus, CR-induced weight loss is the result of a negative energy balance with a potential impact on male reproductive potential [20].

With this work, we aimed to investigate the potential impact of negative energy balance on male reproductive function through the assessment of the physiological/hormonal alterations, oxidative state, and obesity-related genes (ORGs) expression on the testes and spermatozoa. Herein, we focused on four ORGs: fat mass and obesity-associated (*FTO*), melanocortin-4 receptor (*MC4R*), glucosamine-6-phosphate deaminase 2 (*GNPDA2*), and transmembrane protein 18 (*TMEM18*). These ORGs’ expression has already been reported to be altered in human Sertoli cells following hormonal stimulation [21]. In addition, their abundance has previously been associated with sperm quality and embryo development [22]. To achieve our aims of assessing the impact of a negative energy balance and signals on males’ reproductive potential, we used animal models of CR and continuous native GLP-1 administration to evaluate ORGs expression in testes and sperm and how these correlated with sperm quality parameters.

## 2. Materials and Methods

### 2.1. Animal Model and Experimental Design

Sixteen 6-week-old adult male Wistar rats (150–200 g) (Charles River Laboratories, Barcelona, Spain) were housed in our accredited animal colony and were maintained at a constant room temperature (20 ± 2 °C) on a 12 h cycle of artificial lighting with access to food and water. All animal experiments were performed according to the Animal Research: Reporting of In Vivo Experiments Guidelines and were licensed by the Portuguese Veterinarian and Food Department (0421/000/000/2016). After 1 week of acclimatization, body-weight-matched rats were randomly divided into three groups: 5 were used as controls (CTR); 5 were subjected to the peritoneal implantation of a mini-pump for GLP-1 delivery at a constant rate of 3.5 pmol/min/kg for 28 days; and 6 were subjected to caloric restriction (CR). Rats from the CTR and GLP-1 groups were fed ad libitum with a standard chow diet (4RF21 certificate, Mucedola, Italy), whereas the rats in the CR group received 30% less chow diet than the former for 28 days. Food consumption and animal weight were monitored throughout the study duration. Then, the animals were deeply anesthetized with carbon dioxide and were euthanized by exsanguination after a cardiac puncture. Blood was collected into chilled EDTA tubes and was immediately centrifuged, and the plasma was stored at −20 °C until further analysis. The testes were removed and processed for the different procedures and were stored at −80 °C until further use. Epididymides were isolated and placed in prewarmed (37 °C) Hank’s Balanced solution (pH 7.4), crushed, and incubated for 5 min (37 °C). Sperm quality parameters were assessed. Afterward, the suspension was stored at −80 °C until further use.

### 2.2. Assessment of Sperm Quality Parameters

Sperm cells were retrieved from the epididymides according to [20]. The suspension was centrifuged at 300× *g*, 5 min at 37 °C. The spermatozoa were resuspended in PBS. An eosin–nigrosin solution was added to the suspension of spermatozoa (1:2) and smears were performed on microscope glass slides. The smears were left to dry at room temperature. Non-viable spermatozoa were stained pink, whereas viable spermatozoa appeared in white. A total of 100 spermatozoa were counted under a light microscope. The percentage of vitality was then calculated. Smears of the spermatozoa suspension were also performed for a morphology assessment. Once dry, the spermatozoa smears were stained with a Diff-Quick staining kit (Baxter Dale Diagnostics AG, Dubinger, Switzerland) according to the manufacturer’s instructions. A total of 100 spermatozoa were counted under a light microscope. Sperm cells with hook-shaped heads and no defects of the head, neck, or tail were classified as normal. Otherwise, spermatozoa were considered abnormal. The percentage of normal morphology and specific sperm defects was then calculated. The epididymal sperm concentration was determined using a Neubauer counting chamber. After the sperm quality assessment, the remaining spermatozoa were stored at −80 °C.

### 2.3. Assessment of Blood Glucose and Hormonal Levels

Blood glucose levels were determined with the glucose oxidase method using a One Touch Ultra glucometer (Lifescan, Milpitas, CA, USA). Plasma total and active GLP-1 levels (EZGLP1T-36K and EGLP-35K; Merck Millipore, Billerica, MA, USA), insulin (EZRMI-13K; Merck Millipore), total ghrelin (EZRGRT-91K; Merck Millipore, Billerica, MA, USA), and leptin (EZRL-83K; Merck Millipore, Billerica, MA, USA) were determined with an enzyme-linked immunosorbent assay (ELISA) using specific commercial kits according to the manufacturer’s instructions. Insulin resistance was calculated using the homeostatic model assessment for insulin resistance (HOMA-IR) [23].

### 2.4. Polymerase Chain Reaction (PCR) Conditions

To extract the total RNA from the rats’ testes and sperm, the NYZ Total RNA Isolation kit (MB13402; NZYTech, Lisbon, Portugal) was used according to the manufacturer’s instructions. RNA concentration and purity were measured using the microplate reader SYNERGY H1 (BioTek Instruments, VT, USA) at 260/280 nm. Samples with absorbance ratios between 1.8 and 2.1 were considered to have an acceptable quality. Reverse transcription of the total RNA was performed to synthesize cDNA using the NYZ M-MuLV First-Strand cDNA Synthesis Kit (MB17201; NZYTech, Lisbon, Portugal), according to the manufacturer’s instructions, in a final volume of 20 µL. Specific cDNA fragments were amplified using a designed exon–exon spanning primer set. The mRNA levels of the ORGs of interest (*FTO, MC4R, GNPDA2,* and *TMEM18*) and the nuclear factor erythroid 2 like 2 (*NFE2L2)* gene were evaluated in rat testes. The target gene sequences, and annealing temperatures are presented in Table 1. The efficiency of the amplification and quantitative PCR (qPCR) experiments were carried out in duplicate, and the optical density was assessed by a CFX Connect™ Real-Time PCR Detection System (CFX Maestro^TM^ Software 1.1, Bio-Rad, Hercules, CA, USA). The relative gene expression was calculated using the 2^−-∆∆Ct^ method [24] and was normalized to β2-microglobulin (*β2M*) transcript levels.

### 2.5. Immunohistochemistry (IHC)

For protein identification on rats’ testes, we used distinct antibodies with specific concentrations (Table 2). Briefly, 3 μM formalin-fixed paraffin-embedded tissue sections mounted on adhesive microscope slides were deparaffinized and rehydrated in two changes of xylene (10 min each) and 5 min in graded alcohols. Then, tissue sections were subjected, for 8 min, to antigen retrieval through heating by microwaving at 900 W in a 10 mM citrate buffer (pH 6.0), with 0.1% Tween 20 in cases where permeabilization was required. The endogenous peroxidase activity was blocked with a diluted solution of 0.3% hydrogen peroxide in methanol for 15 min. After washing, the slides were incubated in the adequate normal serum (1:5 in 10% bovine serum albumin [BSA]) for 20 min in a humidity chamber. The incubation with the primary antibodies was performed overnight at 4 °C. For reaction detection, the slides were incubated for 60 min with the adequate secondary biotinylated antibody. Afterward, the slides were incubated with the avidin–biotin complex (ABC) (1:100 dilution in 5% BSA; VectorLaboratories, Burlingame, USA) for 30 min, and then, after washing, the reaction was revealed with the DAB substrate (Liquid DAB + Substrate Chromogen System, K3468, Dako, Hamburg, Germany). The sections were washed in distilled H_2_O, counterstained with a 1:2 diluted Harris Hematoxylin, and rinsed in running tap water for 5 min. Then, the sections were dehydrated through a graded series of alcohols for 5 min each and were cleared in two changes of xylene for 5 min each. The immunohistochemistry-stained slides were imaged with a Microscope Digital Camera (Olympus EP50, Tokyo, Japan).

### 2.6. Ferric-Reducing Antioxidant Power Assay (FRAP)

Testicular tissue and sperm were washed with PBS and were centrifuged (8000× *g*, 10 min at 4 °C). The pellets were then separated, diluted in an adequate volume of 1% SDS Buffer, and homogenized. The samples were let to rest for 20 min at 4 °C and then were centrifuged at 14000× *g*, 4 °C for 20 min. The pellet of cellular debris was discarded, and the protein concentration of the supernatant was quantified with a Pierce Bicinchoninic acid protein assay kit (Thermo Fisher Scientific, Massachusetts, USA) using the microplate reader SYNERGY H1 (BioTek Instruments, Vermont, EUA). The total antioxidant potential was determined against standards of L-ascorbic acid by following the absorbance changes at 595 nm. The absorbance results were corrected by using water as the blank. 

### 2.7. Statistical Analysis

All the data presented are expressed as mean ± standard error of the mean (SEM) unless otherwise specified. The Kolmogorov–Smirnov test was used to determine the normality of the groups. A comparison of independent groups was carried out by using either an unpaired t-test or the Mann–Whitney U test. For multiple comparison analysis, the one-way analysis of variance (ANOVA) test was used, with either an ordinary one-way ANOVA or Kruskal–Wallis test depending on the normality of the groups. A *p*-value < 0.05 was considered statistically significant. All the correlations were evaluated by computing Pearson correlation coefficients (*r*) assuming a Gaussian distribution and a confidence interval of 95%. A *p*-value < 0.05 was considered statistically significant. A statistical analysis was carried out using GraphPad Prism (GraphPad Software, Version 8.0.1 for Windows, San Diego, CA, USA).

## 3. Results

### 3.1. Caloric Restriction Decreased Body Weight Gain While No Difference in Body Weight Gain Was Detected after GLP-1 Administration

CR and GLP-1 analogs are known to induce a negative energy balance and decreased body weight. As expected, all rats gained body weight along the experimental period (Figure 1a). However, body weight gain was significantly lower in the CR group as compared to the control group (Figure 1b), while no significant difference was observed between the GLP-1 administered rats and controls (Figure 1b). As expected, the food intake for the control and GLP-1 administration group was superior to the food intake index of the CR group (Figure 1c).

### 3.2. Caloric Restriction Resulted in Lower Insulin Resistance and Altered Adiposity Hormone Profile

There were no significant differences in fasting blood glucose levels across the experimental groups (Table 3). 

CR rats presented significantly lower fasting insulin plasma levels (2.0 ± 0.2 ng/mL vs. 5.6 ± 0.5 ng/mL and 12.3 ± 0.9 vs. 35.3 ± 2.4, respectively) and HOMA-IR when compared to the rats from the control group. Fasting ghrelin plasma levels were significantly higher while fasting leptin plasma levels were significantly lower in rats of the CR group when compared with those of the control group (Table 3).

The GLP-1-administered rats presented no significant differences in fasting insulin plasma levels, HOMA-IR, ghrelin, or leptin when compared to those of the rats from the control group (Table 3). 

Of notice, the GLP-1 administered rats presented significantly higher circulating levels of active GLP-1 when compared to those in the control group (637.3 ± 178.3% vs. 100.0 ± 9.5%, values expressed on % of active GLP-1 in the control group) (Table 3).

### 3.3. GLP-1 Administration Resulted in a Lower Percentage of Sperm Morphology Defects

To evaluate the impact of GLP-1 administration and CR on the reproductive function of male rats, the testes and epididymis were collected. Physiological data regarding the collected organs (weight and gonadosomatic index) and sperm quality parameters can be consulted in Table 4. Neither GLP-1 administration nor CR resulted in significant differences in the testes’ weight, gonadosomatic index, or sperm concentration when compared to the control group. However, significant differences in sperm morphology were detected across the different experimental groups. The GLP-1-administered rats presented a significantly higher percentage of sperm with a normal morphology when compared with rats from the control group (66.21 ± 2.06% vs. 60.98 ± 2.31%). The CR rats showed a significantly higher percentage of sperm head defects compared to rats in the control group (9.4 ± 0.8% vs. 5.7 ± 1.3%), as well as a higher percentage of non-viable sperm.

### 3.4. Identification of ORG Transcripts and Protein on the Testes of Wistar Rats

Several ORGs encode proteins involved in the leptin–melanocortin signaling pathway. This pathway is important in controlling the energy homeostasis in the hypothalamus by coordinating the appetite, food intake, and energy expenditure. Thus, we identified the presence of the four selected ORGs transcripts in both rat testes and sperm with conventional PCR (Figure 2). We confirmed the presence of *FTO*, *MC4R,* and *TMEM18* transcripts in rat testes, as reported in previous works [25,26,27]. To our knowledge, this is the first time that *FTO* has been identified in rat sperm, as represented in Figure 2a.

Once we identified the presence of ORGs transcripts in the rat testes and sperm, we further proceeded to investigate the ORGs protein expression and location on rat testes via immunohistochemistry (Figure 3). We were able to identify the presence of all the studied ORGs proteins in the rat testicular tissue. FTO was present in rat testes (Figure 3a), specifically in primary spermatocytes, spermatogonia, and Leydig cells. MC4R was highly expressed in the primary spermatocytes, spermatogonia, myoid cells, and Leydig cells (Figure 3b). GNPDA2 appeared to be present in late elongated spermatids and Leydig cells (Figure 3c). TMEM18 was expressed in the early undifferentiated spermatogonia (Figure 3d).

### 3.5. Caloric Restriction and GLP-1 Administration Increased ORG Expression in Testes

Energy status can influence gene (mRNA levels) expression [28]. Interventions such as GLP-1 analogs administration and CR can lead to a negative energy balance status [29,30]. Thus, we hypothesized that in the high energy demanding testes and sperm, the expression of the selected ORGs, involved in the body’s energy homeostasis, might respond to those interventions.

Indeed, we observed a higher expression of *FTO* (3.43 ± 0.64 vs. 1.01 ± 0.06 (Figure 4a)), *MC4R* (9.03 ± 0.61 vs. 1.00 ± 0.04 (Figure 4b)), *GNPDA2* (8.27 ± 0.53 vs. 1.02 ± 0.09 (Figure 4c)), and *TMEM18* (10.52 ± 3.75 vs. 1.04 ± 2.01 (Figure 4d)) in the testes of GLP-1-administered rats in comparison to the control. A higher *FTO, MC4R, GNPDA2,* and *TMEM18* expression in CR rat testes when compared to the control group was also observed (5.30 ± 1.75 vs. 1.01 ± 0.06 (Figure 4a), 4.77 ± 1.37 vs. 1.00 ± 0.04 (Figure 4b), 2.89 ± 0.69 vs. 1.02 ± 0.09 (Figure 4c), and 8.87 ± 3.75 vs. 1.04 ± 0.16 (Figure 4d), respectively).

Neither GLP-1 administration nor CR resulted in any significant difference in the abundance of ORGs transcripts in rat sperm when compared to rats from the control group (Figure 5).

### 3.6. CR and GLP-1 Administration Increased Testicular NEF2L2 Expression but Did Not Alter Testes’ Total Antioxidant Capacity

Oxidative stress responds to metabolic and endocrine cues. Oxidative stress can result from a decrease in the antioxidant capacity or an increase in reactive oxygen species (ROS) production [31]. Rats’ testes total antioxidant capacity (TAC), assessed using the FRAP assay was not significantly different across the study groups (Figure 6a). Our data also showed that TAC in CR rat sperm was not significantly different when compared to those of the rats of the control group (Figure 6b). Nrf2, encoded by the nuclear factor erythroid 2 like 2 (*NFE2L2*) gene, is a transcription factor that regulates the cellular defense against toxic and oxidative insults through the regulation of the expression of the genes involved in the oxidative stress response. Further, in a previous study, its expression has been correlated with the expression of *FTO* [25].

We confirmed the presence of the *NFE2L2* transcript in both rat testes and sperm through conventional PCR (Figure 7a), as reported by Wadja and colleagues [32]. We also observed that the expression of *NFE2L2* was higher in the testes of the GLP-1-administered and CR rats when compared to those of the control group (7.15 ± 1.98 vs. 1.11 ± 0.23 and 13.77 ± 2.96 vs. 1.11 ± 0.23, respectively) (Figure 7b). On the other hand, there were no significant differences in *NFE2L2* expression in sperm from rats across the study groups (Figure 7c).

## 4. Discussion

The male reproductive function depends on energy homeostasis [33]. Several single nucleotide polymorphisms (SNPs) of the genes involved in metabolism and energy balance regulation were identified as being highly associated with obesity and that is why they are known as obesity-related genes—ORGs [34]. Most of these SNPs are in introns or intergenic regions, suggesting that these affect the regulation of the corresponding gene or nearby genes [35]. Many genes involved in energy homeostasis are regulated in response to feeding and fasting or dietary components [36]. To further understand the regulation of ORGs’ expression in testes and sperm, we studied how interventions known to interfere with energy balance affect their expression. Herein, we studied whether the selected ORGs—*FTO*, *MC4R*, *GNPDA2*, and *TMEM18*—were expressed in rat testes and sperm and how their expression responds to energy restriction or a signal of a negative energy balance, namely CR and GLP-1 administration. In addition, we determined if those interventions affected sperm quality with a potential impact on male reproductive function.

CR promotes a negative energy balance to achieve weight loss [16]. CR rats presented a lower body weight gain, lower fasting insulin, and lower HOMA-IR value when compared to controls, compatible with decreased insulin resistance. CR also led to lower leptin levels, a finding compatible with lower adiposity, while ghrelin levels were higher, suggesting energy deprivation. Overall, the biometric parameters and hormone profile observed in the CR rats highlight the effectiveness of this dietary intervention in inducing a negative energy balance signaling. GLP-1 is a gastrointestinal hormone that has been demonstrated to decrease food intake and promote weight loss in a dose-dependent manner, both in humans and animals [37,38]. Continuous intraperitoneal GLP-1 administration in rats, despite eliciting significantly higher levels of active GLP-1 and an active/total GLP-1 ratio when compared to controls had no significant impact on food intake or body weight gain. Likewise, fasting glucose, insulin, and HOMA-IR, as well as leptin and ghrelin were unsurprisingly not significantly different when compared to those in the control group. The hormone profile observed after GLP-1 administration confirms the effectiveness of this intervention in enhancing GLP-1 bioavailability as a negative energy balance signal, although not to the extent of influencing food intake, systemic energy status, or glucose homeostasis.

Genes such as *FTO, MC4R, GNPDA2,* and *TMEM18* are known to encode the factors that are involved in food intake and energy expenditure regulation [39]. As energy balance is known to modulate spermatogenesis, we hypothesized that CR and GLP-1 administration could impact male reproductive function by altering sperm quality, and this effect could be mediated or associated with the ORGs’ expression.

*FTO* is highly expressed in brain tissue and controls eating behavior, having a role in body mass regulation. However, the role of FTO in other tissues, including testes, is still not clear. Our study confirmed that FTO is expressed in rat testes, as previously reported [25], and herein reports its expression in rat spermatozoa for the first time. We were also able to demonstrate that the expression of the correspondent protein in the testes is mainly located in primary spermatocytes, spermatogonia, and Leydig cells. How *FTO* is affected by nutritional status is somewhat controversial, namely in the hypothalamus, which has been the main studied tissue. Most of the studies found an increased *FTO* gene expression after CR [40,41,42], in agreement with our results, but for example, Poritsano and colleagues [43] claimed a decreased *FTO* expression after CR. To our knowledge, there are no other studies that investigated the effect of CR or the effect of GLP-1 administration on the expression of *FTO* in testes and sperm. In this follow up, we demonstrated that *FTO* expression in testes was higher in the rats subjected to CR and GLP-1 administration, but it remained unchanged in sperm. We hypothesized that the lower oxidative stress-induced damage observed in the testes after CR, demonstrated by Martins and colleagues [20] using this same animal model, could be associated with the higher *FTO* expression observed in the rat testes. CR is known to promote a decrease in oxidative damage in several tissues and organs [44]. The proposed relationship could be mediated by an increase in the *NFE2L2* gene expression that encodes for the Nrf2 protein which is, in turn, an important transcription factor of antioxidant defense-related genes. We raise this hypothesis by also taking into consideration what Zhao and colleagues [25] observed in a study related to the effects of the environmental endocrine-disrupting chemical di-(2-Ethylhexyl) phthalate (DEPH). They observed that this compound promoted oxidative stress-induced testicular damage and decreased *FTO* expression. These two aspects were related since the decreased *FTO* expression led to an increased global level of m6A RNA modification, particularly in *Nrf2* mRNA. This occurred because *FTO* is an RNA methylation modulator gene responsible for removing specific methylations. The altered m6A modification has an important role in male reproductive dysfunction, so in this case, the altered m6A modification in Nrf2 mRNA leads to lower Nrf2 protein levels, affecting the Nrf2-mediated antioxidant system and leading to aggravated oxidative stress and testicular injury [25]. In agreement with our hypothesis, we found that the higher *FTO* expression was correlated with greater *NFE2L2* expression in the testes (Appendix A). That positive correlation was also observed in sperm, although there were no significant differences in the expression of *FTO* and *NFE2L2* in the CR group when compared to the control. Interestingly, *FTO* and *NFE2L2* expression in the testes was positively correlated with sperm total antioxidant capacity. In addition, sperm *FTO* expression was positively correlated with sperm concentration (Appendix A). A positive correlation between sperm *FTO* abundance and the total sperm count in humans was also reported by Pereira SC and colleagues [22]. This suggests that CR may indirectly induce the Nrf2-mediated antioxidant pathway in rat testes and consequently decrease the oxidative stress through increased *FTO* expression. This may be associated with sperm quality, namely with sperm concentration, which despite presenting no significant differences when compared to the control group showed a positive correlation with two genes that seem to be linked to the control of oxidative stress in the testes. Thus, the *FTO* and *NFE2L2* genes in the sperm represent potential targets for increasing sperm concentration.

*MC4R* encodes for a receptor that belongs to the melanocortin receptor family, which has a central role in energy homeostasis regulation [45]. Its roles in the central nervous system are quite well characterized, yet its potential role in the male reproductive system remains unknown. As reported, we observed that *MC4R* is expressed in rat testes [26], and we also identified its expression in rats’ sperm for the first time. We also showed that the expression of this protein in the testes was mainly located in primary spermatocytes, spermatogonia, myoid cells, and Leydig cells. Furthermore, we demonstrated that its expression in the testes was increased in the rats subjected to CR and GLP-1 administration. The *MC4R* expression in the testes and sperm of the CR rats was positively correlated with *FTO* expression in the testes, and *FTO* was associated with the oxidative status of the rat testes (Appendix A). Indeed, the expression of *MC4R* in sperm was found to be positively correlated with the total antioxidant capacity and with the expression of *NFE2L2* in testes. In the rats subjected to GLP-1 administration, we also observed a strong positive correlation between *MC4R* and *NFE2L2 expressions* in sperm (Appendix A). This leads us to suggest that *MC4R,* in rats subjected to CR and GLP-1 administration, similarly to what was observed in *FTO*, has an active role in protecting against testicular oxidative damage, especially in sperm.

*GNPDA2* encodes an allosteric enzyme that catalyzes the deamination of glucosamine-6-phosphate (GlcN6P) into fructose-6-phosphate (F6P) in the hexosamine signaling pathway (HSP), which is one of the main nutrient-sensing pathways [46]. Our study reported for the first time that *GNPDA2* is expressed in rat testes and sperm. We also showed that the expression of its protein in the testes was mainly located in late spermatids and Leydig cells. Its expression was higher in the testes of rats subjected to CR and GLP-1 administration. Glycosylation, which is the enzymatic process of attaching glycans or carbohydrates to proteins, lipids, or other organic molecules, is the main reaction and is indispensable namely in the process of spermatogenesis. The glycosylation of sperm occurs during spermatogenesis and the maturation process occurs during epididymal transit and capacitation [47]. Relative to sperm quality, the sperm of the CR rats presented a significantly higher percentage of head defects compared to those of the control group. The higher percentage of sperm head defects is related to lower fertility potential [48]. Indeed, a higher percentage of sperm head defects are present in infertile individuals [48], which is also related to low pregnancy rates [49]. We noticed a positive correlation between these head defects and *GNPDA2* expression in the testes and sperm from CR rats (Appendix A), which leads us to hypothesize that these effects are somehow related to CR impact on *GNPDA2* expression, namely due to an associated decrease in UDP-GlcN6P and consequently O-glycosylation. In humans, higher levels of *GNPDA2* transcript in sperm cells were also associated with decreased sperm quality [22]. A study recently showed that the sperm from highly fertile bulls possessed a higher abundance of O-linked glycans than the lower fertile bulls [50]. Thus, the effect of CR on *GNPDA2* seems to reverse the HSP flow in the opposite direction to the formation of UDP-GlcN6P. This process is beneficial by decreasing insulin resistance. Regardless, it is associated with lower sperm quality, namely by the increase in head defects. In rats subjected to GLP-1 administration, we noticed that despite not presenting significant phenotypic or systemic differences related to energy balance when compared to the control group, the expression of *GNPDA2* in sperm was positively correlated with the expression of *NFE2L2*. In addition, there was also a positive correlation between *GNPDA2* expression in sperm and sperm concentration (Appendix A). Thus, in sperm, *GNPDA2* may play a role in controlling the oxidative status through the Nrf2-mediated antioxidant pathway that eventually positively impacts sperm concentration.

The *TMEM18* gene encodes a three-transmembrane domain protein with a positive-charge C-terminus domain and its function is still under debate, with the information concerning its role in the male reproductive function being practically null [27]. Our study confirmed that *TMEM18* is expressed in rat testes, as reported in [27], and we reported its expression in rat sperm for the first time. We were also able to show that the expression of its protein in the testes is mainly located in early spermatogonia. Finally, we demonstrated that its expression in the testes was increased in rats subjected to CR and GLP-1 administration. A positive correlation between *TMEM18* and peroxisome proliferator-activated receptor gamma (*PPARG)* in human adipose tissue was previously described. *TMEM18* affects adipogenesis via a direct or indirect functional interaction with *PPARG* [51]. Thus, we hypothesize that *TMEM18* function in the testes might be mediated by PPARG. PPARG has the potential to improve insulin resistance since its activation increases peripheral tissue sensitivity to insulin [52,53]. We observed a negative correlation between HOMA-IR, which was significantly lower in the CR group, and the unchanged *TMEM18* expression in the sperm of these rats (Appendix A). Insulin resistance negatively affects sperm quantity and quality [52]. We also found a positive correlation between *TMEM18* expression in the sperm of CR rats and its sperm concentration (Appendix A). Although the expression of sperm *TMEM18* and sperm concentration did not differ in the CR rats as compared to those of the control group, this suggests that the association of *TMEM18* expression with sperm concentration could be mediated by the impact in insulin resistance. Furthermore, a study demonstrated that prostaglandin J2, an agonist of PPARG, increases the viability of sperm, whereas all these events are irreversibly reduced by the PPARG antagonist GW9662, confirming the involvement of PPARG in sperm viability [52]. Since *TMEM18* and PPARG expressions are closely linked, we observed that *TMEM18* expression in the testes of the CR rats had a positive correlation with sperm vitality (Appendix A). That correlation was not observed in the control group. Thus, it suggests that despite the fact that the sperm vitality of the CR rats did not show a significant difference when compared to the control rats, the expression of *TMEM18* in the former becomes associated with a higher sperm vitality.

There is a reciprocal transcriptional regulation between the genes of *Nrf2* and *PPARG*. A positive feedback loop between *PPARG* and *Nrf2* seems to exist, being that *PPARG* may act directly or through an upstream pathway for Nrf2 activation. Nuclear receptor *PPARG* and transcription factor *Nrf2* may act synergically in the activation of antioxidant defense genes [53]. In this study, we observed that *TMEM18* expression in the testes of rats subjected to CR had a positive correlation with the *NFE2L2* expression also in its testes (Appendix A), being that both expressions were significantly higher in the animals of the CR group when compared to the control group. This same correlation also occurred in the sperm of these rats, but in this case, the expression of both *TMEM18* and *NFE2L2* had no significant differences compared to the control group. Furthermore, we detected that the expression of *TMEM18* in the CR rats’ testes was positively correlated with sperm total antioxidant capacity. This leads us to hypothesize that *TMEM18* may have a beneficial role in the oxidative state of both the testes and sperm of the rats subjected to CR, through the activation of the Nrf2-mediated antioxidant system. Interestingly, we also observed that there was a positive correlation between the expression of *TMEM18* and *FTO* in both the testes and sperm of rats subjected to CR, being that *FTO* was also linked to a decrease in oxidative stress, especially in the testes (Appendix A). Similar to what was observed in the CR group, the expression of *TMEM18* in sperm from the rats subjected to GLP-1 administration was positively correlated with the expression of *NFE2L2* also in sperm (Appendix A). This leads us to hypothesize that the administration of GLP-1 may promote a greater antioxidant capacity through its impact on *TMEM18* and consequent activation of the Nrf2-mediated antioxidant system.

## 5. Conclusions

In this study, we described the impact of energy deprivation, achieved through CR and conveying a negative energy balance signal through native GLP-1 administration, on sperm quality.

CR resulted in lower body weight gain and insulin resistance, but a higher percentage of sperm head defects. GLP-1 administration, despite showing no influence on body weight or glucose homeostasis, resulted in a lower percentage of sperm head defects. Both CR and GLP-1 administration increased ORGs expression in the testes, but not in spermatozoa. In CR rats, *FTO* was positively correlated with the sperm’s total antioxidant capacity, concentration, and vitality; *MC4R* also appeared to be linked with the improved oxidative status of the testes and sperm. The effect of CR on *GNPDA2* seems to reverse the HSP flow in the opposite direction to the formation of UDP-GlcN6P. This process could be beneficial by decreasing insulin resistance. Regardless, the quick metabolic alterations promoted by CR promote an increase in the percentage of sperm head defects. This is likely caused by the decrease in glycosylation, which is essential for spermatogenesis and sperm maturation. Finally, *TMEM18* was positively associated with concentration and sperm viability under CR conditions. Furthermore, it seems to promote improvements related to oxidative stress in the testes and the total antioxidant capacity of sperm. Overall, our results suggest an association between CR, the increase in ORGs expression, and the improvement in the oxidative status in the testes.

GLP-1 administration seems to have beneficial effects on male reproductive potential beyond the effects on body weight. After GLP-1 administration, although ORGs expression in the sperm was not significantly altered, the expression of *MC4R*, *GNPDA2,* and *TMEM18* was strongly positively correlated with *NFE2L2* expression in sperm. The sperm *GNPDA2* expression was also positively correlated with sperm concentration. Thus, future studies should focus on assessing whether GLP-1 analogs can replicate or even potentiate the effects of native GLP-1 in sperm quality parameters as shown in this study.

## Figures and Tables

**Figure 1 biomedicines-10-02609-f001:**
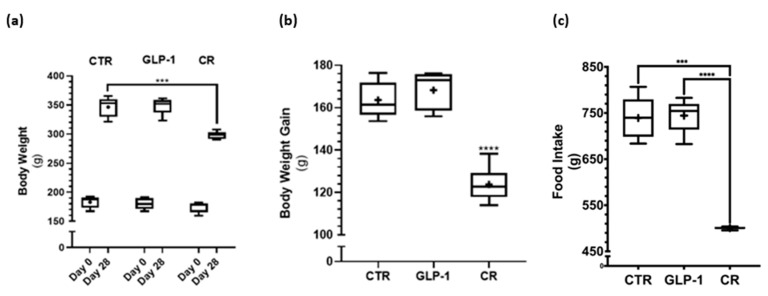
Body weight response to 28 days of glucagon-like peptide-1 (GLP-1) administration and caloric restriction (CR). (**a**) The body weight of the rats from the control (CTR), GLP-1, and CR groups was measured on day 0 and day 28 of the experiment. (**b**) The body weight gain of the rats from the CTR, GLP-1, and CR groups was calculated considering the difference in weight between day 28 and day 0 of the experiment. (**c**) The food intake of the rats from the CTR, GLP-1, and CR groups. Results are expressed as mean ± SEM (n = 5 for CTR and GLP-1 condition and n = 6 to CR condition). * indicates statistically significant differences as compared to the CTR group (*** *p* < 0.001 and **** *p* < 0.0001). + indicates mean value.

**Figure 2 biomedicines-10-02609-f002:**
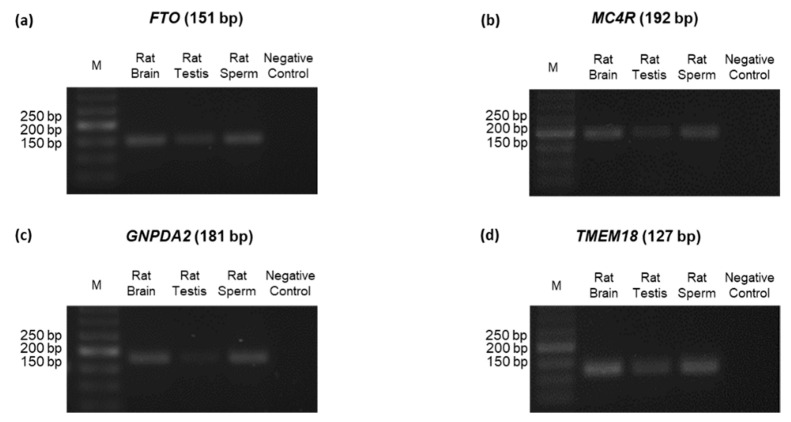
Identification of ORGs transcripts in rats’ testes and sperm by conventional PCR. Fat mass and obesity-associated (*FTO*) (**a**), melanocortin-4 receptor (*MC4R*) (**b**), glucosamine-6-phosphate deaminase 2 (*GNPDA2*) (**c**), and transmembrane protein 18 (*TMEM18*) (**d**) transcripts in both Wistar rat testes and sperm were identified with conventional PCR. A cDNA-free vehicle was used as a negative control and a rat brain was used as a positive control.

**Figure 3 biomedicines-10-02609-f003:**
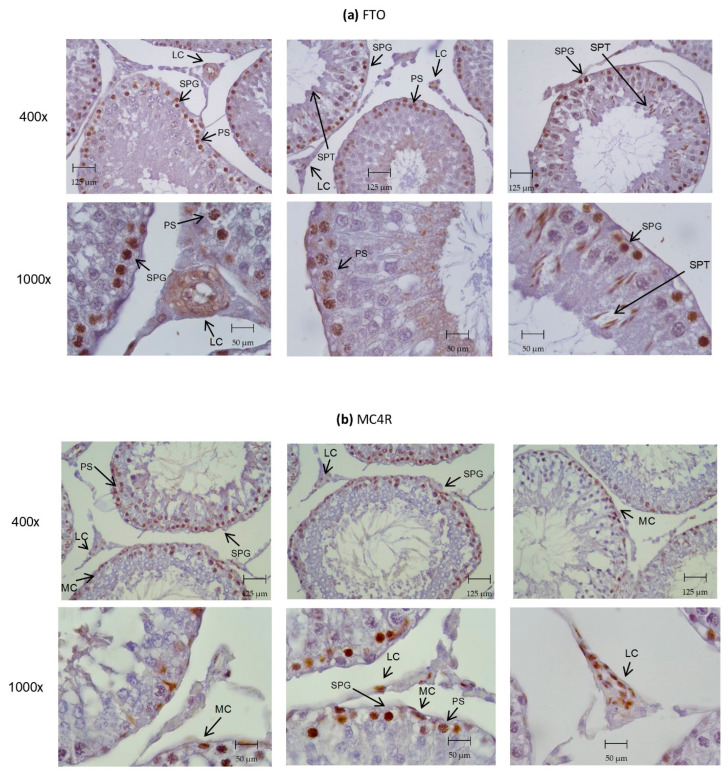
Immunohistochemistry staining for the studied ORGs in rats’ testes. Immunohistochemical staining for the expression of FTO (**a**), MC4R (**b**), GNPDA2 (**c**), and TMEM18 (**d**) in rats’ testes. The images were obtained in two different magnifications: 10× (eyepiece) × 40× (objective) and 10× (eyepiece) × 100× (objective). Abbreviations: LC, Leydig cell; MC, myoid cells; PS, primary spermatocyte; SC, Sertoli cell; SPC, spermatocyte; SPG, spermatogonia; SPT, spermatid.

**Figure 4 biomedicines-10-02609-f004:**
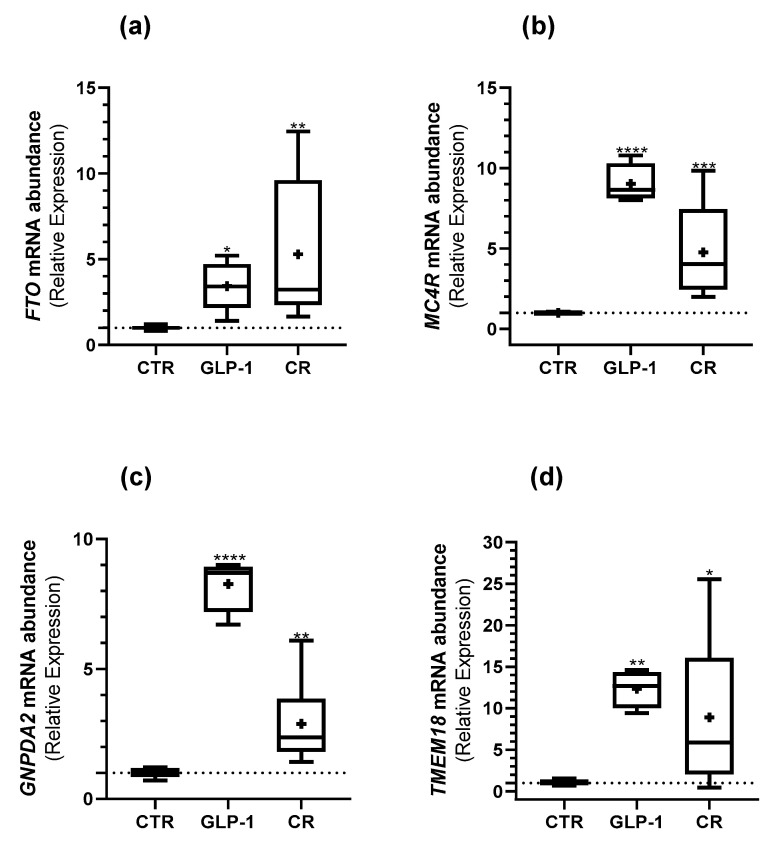
Effect of GLP-1 administration and CR on the studied ORGs expression in rats’ testes. The expression of ORGs (FTO (**a**); MC4R (**b**); GNPDA2 (**c**); TMEM18 (**d**)) in rat testes was assessed with qPCR. Results are expressed as mean ± SEM (n = 5 for CTR and GLP-1 condition and n = 6 to CR condition). * indicates statistically significant differences as compared to the CTR group (* *p* < 0.05, ** *p* < 0.01, *** *p* < 0.001, and **** *p* < 0.0001). + indicates mean value.

**Figure 5 biomedicines-10-02609-f005:**
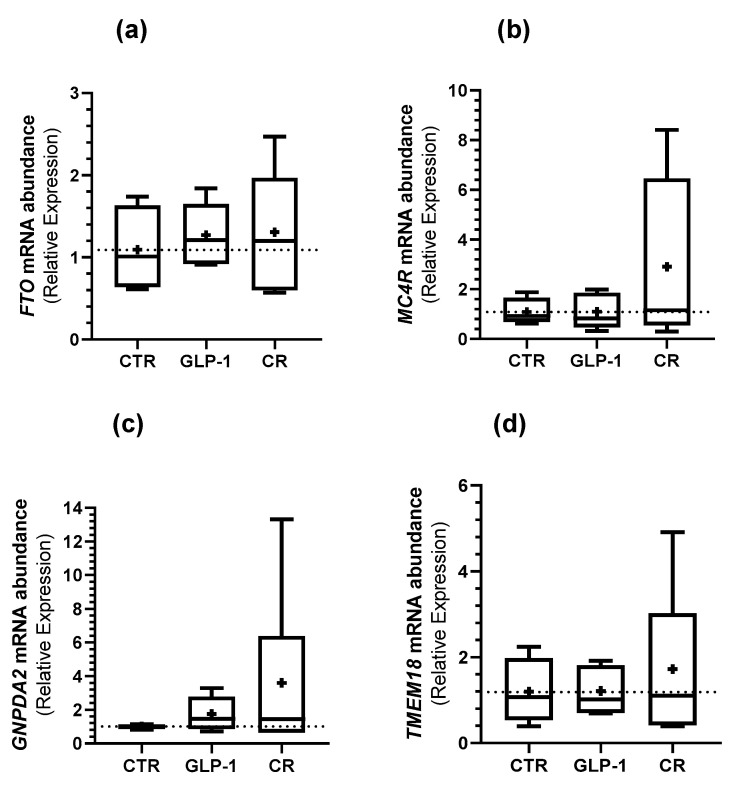
Effect of GLP-1 administration and CR on the studied ORGs expression in rats’ sperm. The expression of ORGs ((**a**) *FTO*; (**b**) *MC4R*; (**c**) *GNPDA2*; (**d**) *TMEM18*) in rats’ sperm was assessed with qPCR. Results are expressed as mean ± SEM (n = 5 for CTR and GLP-1 condition and n = 6 to CR condition). + indicates mean value.

**Figure 6 biomedicines-10-02609-f006:**
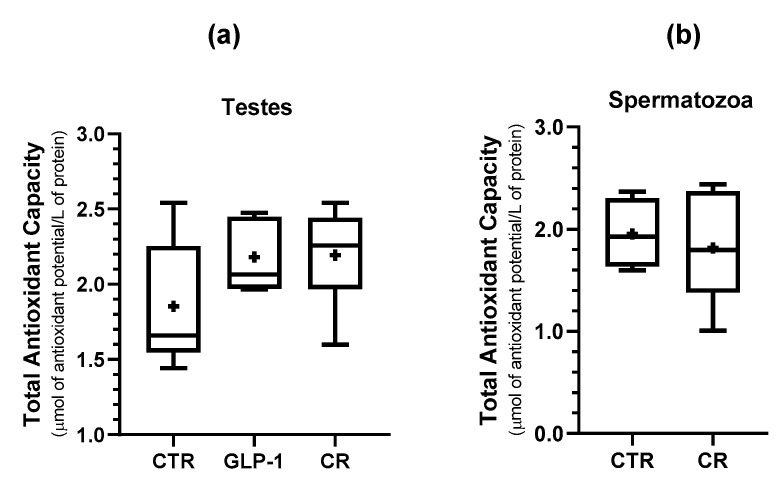
Effect of GLP-1 administration and CR on the total antioxidant capacity of spermatozoa and testes. The Ferric-Reducing Antioxidant Power Assay (FRAP) was measured in the rat testes (**a**) and sperm (**b**). The antioxidant power is expressed by the FRAP value (µmol of antioxidant potential/L of protein) as mean ± SEM (n = 5 for CTR and GLP-1 condition and n = 6 for CR condition). + indicates mean value.

**Figure 7 biomedicines-10-02609-f007:**
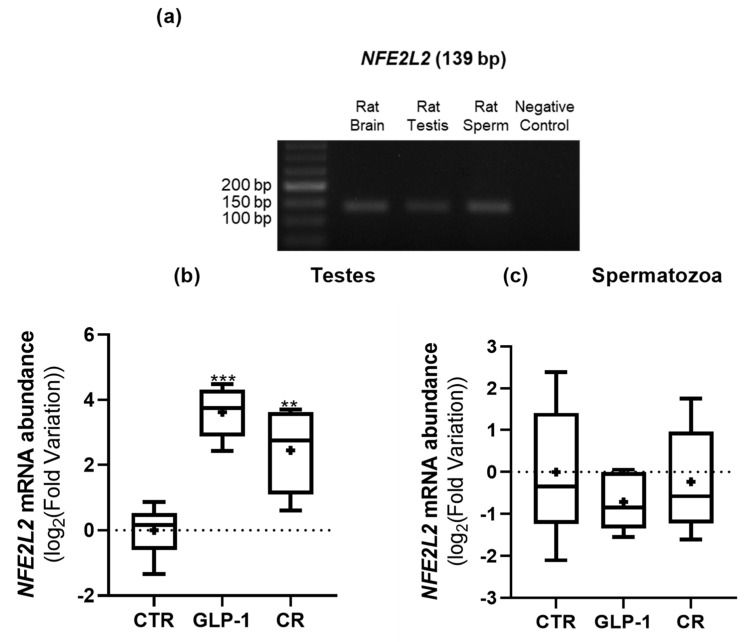
Expression of nuclear factor erythroid 2 like 2 (NFE2L2) gene transcripts in rats’ testes and sperm. (**a**) NFE2L2 transcripts in both Wistar rats’ testes and sperm were identified with conventional PCR. A cDNA-free vehicle was used as a negative control and a rat brain was used as a positive control. The expression of NFE2L2 in rats’ testes (**b**) and sperm (**c**) was assessed with qPCR. Results are expressed as mean ± SEM (n = 5 for CTR and GLP-1 condition and n = 6 to CR condition). * indicates statistically significant differences as compared to the CTR group (** *p* < 0.01 and *** *p* < 0.001. + indicates mean value.

**Table 1 biomedicines-10-02609-t001:** Primer sequences and Polymerase Chain Reaction (PCR) conditions used to assess gene expression and mRNA abundance in rats’ testes and sperm. β2-microglobulin (*β2M*) was used as a housekeeping control.

Gene Name	GenBank Accession Number	Primer sequence (5′–3’)	Amplicon Length	Annealing Temperature	Cycles
*β2M*	NM_012512.1	Forward: CCGTGATCTTTCTGGTGCTTGTC	150 bp	58.0 °C	35
		Reverse: CTATCTGAGGTGGGTGGAACTGAG			
*MC4R*	NM_013099.3	Forward: ACAAGAACCTGCACTCACCC	192 bp	62.0 °C	35
		Reverse: ATGCGAGCAAGGAGCTACAG			
*FTO*	NM_001039713.1	Forward: CAGAGATCCCGATACGTGGC	151 bp	57.6 °C	35
		Reverse: CTGTGAGCCAGCCAAAACAC			
*GNPDA2*	NM_001106005.1	Forward: ACCATCCCGAAAGCTACCAT	181 bp	58.4 °C	35
		Reverse: GGACCTATTCCTCCAACAAAAAGAT			
*TMEM18*	NM_001007748.1	Forward: AAGCATGGTGAATGGGGACC	127 bp	56.4 °C	35
		Reverse: CACACTCAAACCTGCGTGAC			
*NFE2L2*	NM_031789.2	Forward: CAATGACTCTGACTCCGGCA	139 bp	56.6 °C	35
		Reverse: AGGGGCACTGTCTAGCTCTT			

**Table 2 biomedicines-10-02609-t002:** List of antibodies and respective concentrations used for identification and location of the Obesity Related Genes (ORGs) correspondent proteins.

Antibody	Company	Catalog Number	Concentration for IHC
Anti-FTO	Abcam	ab92821	1:300
Anti-MC4R	Abcam	ab24233	1:75
Anti-GNPDA2	Abcam	ab106363	1:150
Anti-TMEM18	Abcam	ab100954	1:150

**Table 3 biomedicines-10-02609-t003:** Fasting blood glucose and hormones levels. Fasting blood glucose and hormone measurements (insulin, ghrelin, active GLP-1, and leptin) were performed in rats from the CTR, GLP-1, and CR groups. Homeostatic Model Assessment for Insulin Resistance (HOMA-IR) value was also calculated. Results are expressed as mean ± SEM (n = 5 for control and GLP-1 condition and n = 6 to CR condition). * indicates statistically significant differences as compared to the control group (* *p* < 0.05, ** *p* < 0.01 and *** *p* < 0.001). ↑ indicates statistically increase in comparison to the control, ↓ indicates statistically decrease in comparison to the control, and ↔ indicates no statistically significant difference in comparison to the control.

Parameter	Control	GLP-1 Administration	Caloric Restriction
Active GLP-1 (pM)	2.5 ± 0.2	15.7 ± 4.4 ↑ **	5.0 ± 1.5 ↔
Glucose (mg/dL)	89.6 ± 2.2	102.6 ± 7.1 ↔	88.0 ± 3.2 ↔
Insulin (ng/mL)	5.6 ± 0.5	5.2 ± 0.9 ↔	2.0 ± 0.2 ↓ ***
HOMA-IR	35.3 ± 2.4	38.8 ± 8.1 ↔	12.3 ± 0.9 ↓ **
Ghrelin (ng/mL)	0.6 ± 0.1	0.6 ± 0.2 ↔	1.0 ± 0.2 ↑ *
Leptin (ng/mL)	6.9 ± 0.7	6.3 ± 0.8 ↔	2.1 ± 0.3 ↓ ***

**Table 4 biomedicines-10-02609-t004:** Impact of GLP-1 administration and CR on testes and epididymis weights, gonadosomatic index, and sperm quality parameters. The different responses of the rats from the CTR, CR, and GLP-1 groups’ reproductive structures was evaluated by the sum of the weights of the right and left testes and epididymis. The gonadosomatic index was also calculated. The sperm quality was evaluated by the measurement of sperm concentration, viability, and morphology. In addition to morphology, sperm defects specifically in the head, neck, and tail were also determined. The results are expressed as mean ± SEM (n = 5 for CTR and GLP-1 condition and n = 6 to CR condition). * indicates statistically significant differences as compared to the CTR group (* *p* < 0.05).

Parameter		Control	GLP-1 Administration	Caloric Restriction
Testes Weight (g)		3.6 ± 0.1	3.3 ± 0.2	3.2 ± 0.2
Epididymis Weight (g)		1.3 ± 0.1	1.1 ± 0.1	1.0 ± 0.1
Gonadosomatic Index (%)		1.0 ± 05	0.9 ± 0.6	1.1 ± 0.8
Sperm Concentration (Spermatozoa/mL × 10^6^)		39.5 ± 4.6	45.1 ± 4.2	30.8 ± 4.7
Sperm Vitality (% Spermatozoa)	Viable	42.6	43.8	36.2
	Non-viable	57.4	56.2	63.8
Sperm Morphology (% Spermatozoa)	Normal	60.6 ± 2.3	66.2 ± 2.1 *	59.5 ± 1.7
	Head Defects	5.7 ± 1.3	5.3 ± 1.0	9.4 ± 0.8 *
	Neck Defects	19.6 ± 3.2	15.9 ± 2.7	19.1 ± 3.3
	Tail Defects	13.7 ± 1.1	12.0 ± 2.2	12.6 ± 2.2

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
