# Peer review of "Obesity-Related Genes Expression in Testes and Sperm Parameters Respond to GLP-1 and Caloric Restriction"

_biomedicines, 2022, doi:10.3390/biomedicines10102609_

Round 1

Reviewer 1 Report

In their manuscript, the authors study the effect of caloric restriction (30%) and GLP-1injections, two conditions linked to a negative energy balance, on metabolic parameters, sperm quality, expression of ORGs and antioxidant capacity in testis and spermatozoa. The paper is clear, easy to read and well-illustrated. The experiments and results are clearly described and convincing.

However, the difficulty is that for me the results do not fully support the conclusions. Although the article's title effectively sums up the results, the discussion and the conclusions seem to over-interpret the results and are too speculative. See for instance: “Conclusions: CR and GLP-1 administration results in higher ORGs expression in testis and these were correlated to an overall improvement of sperm fertility parameters”. Overall, the authors combine the results obtained with CR and GLP-1 into one general conclusion, while the effects on sperm quality are quite different. I'm not sure if it's appropriate to do so. Moreover, I find it problematic that the GLP-1 treatment has no effect on the metabolic parameters measured when the authors want to study the link between energy balance and sperm quality. Moreover, there is no demonstration of the role of the studied genes in changes in sperm quality, although the authors discuss these potential links extensively in their discussion.

More specifically, how to explain that the two conditions studied here induce comparable changes in the expression of ORGs while the effect on sperm quality is different? CR increases the percentage of sperm with defects, while GLP-1 treatment (which has no impact on energy balance) increases the percentage of normal sperm. If a link exists between the modification of the expression of ORGs and the sperm quality, I would have expected comparable results for CR and GLP-1. How do the authors explain this? This point is never addressed in the manuscript. The authors also speculate on a possible modulation of oxidative stress following the treatment, but the total antioxidant capacity is not modified. The highlighted increase in nfr2 expression remains a very indirect clue. As before, if the effect on oxidative capacity is comparable for CR and GLP-1 (increased expression of nfr2), how can the authors reconcile this result with the differential effects of the 2 treatments on sperm quality. Based on the results, the following sentence is incorrect and should be rewritten. "Overall, we were able to unveil that CR is associated with increased ORGs expression and improvement in oxidative status both in tests and sperm." CR does not modify nfr2 in sperm. Regarding the effect of GLP-1, how do the authors envisage its action in the testicles? Is this a direct effect? Are there GLP-1 receptors at this level? The authors should discuss this.

Author Response

First, we would like to acknowledge the Reviewers for the dedicated analysis of the manuscript and the questions addressed to the different subjects, in order to improve the understanding of experimental work and the rationalization of its results. Thereby, we dissected the questions and, point-by-point, we answered them, signalizing in yellow the sentences or sections that were altered in the manuscript.

Reviewer #1

General comment: In their manuscript, the authors study the effect of caloric restriction (30%) and GLP-1 injections, two conditions linked to a negative energy balance, on metabolic parameters, sperm quality, expression of ORGs and antioxidant capacity in testis and spermatozoa. The paper is clear, easy to read and well-illustrated. The experiments and results are clearly described and convincing.

Answer: We thank the reviewer for these encouraging words. We appreciate the fact that, as we envisioned, we were able to carry out this project and later write it in a way that was easily read and interpreted by the scientific community.

Issue 1: However, the difficulty is that for me the results do not fully support the conclusions. Although the article's title effectively sums up the results, the discussion and the conclusions seem to over-interpret the results and are too speculative. See for instance: “Conclusions: CR and GLP-1 administration results in higher ORGs expression in testis and these were correlated to an overall improvement of sperm fertility parameters”. Overall, the authors combine the results obtained with CR and GLP-1 into one general conclusion, while the effects on sperm quality are quite different. I'm not sure if it's appropriate to do so.

Answer 1: Thank you for your suggestion. We agree with the reviewer, and we rewrote the conclusion. In this new version of the manuscript, it can now be read: “Conclusions: CR and GLP-1 administration results in higher ORGs expression in testes, and these were correlated to several alterations on sperm fertility parameters”. Thank you for the suggestion.

Issue 2: Moreover, I find it problematic that the GLP-1 treatment has no effect on the metabolic parameters measured when the authors want to study the link between energy balance and sperm quality.

Answer 2: We thank the reviewer for this question. We can explain that with the presumption that the lack of impact of GLP-1 on metabolic parameters is not surprising, given that our animal model includes normal-weight, non-diabetic rats. In human patients, the effects of GLP-1 can only be observed in patients with excess weight/obesity or other metabolic abnormal condition. Once the metabolism is normalized, no effects regarding the administration of GLP-1 (analogs) can be observed, despite chronic therapy. For this work, we choose to work with normal weight, no diabetic animals, because this makes it possible to evaluate the effect of GLP-1 a mediator of energy regulation, independently of weight or metabolic changes.

Issue 3: Moreover, there is no demonstration of the role of the studied genes in changes in sperm quality, although the authors discuss these potential links extensively in their discussion. More specifically, how to explain that the two conditions studied here induce comparable changes in the expression of ORGs while the effect on sperm quality is different? CR increases the percentage of sperm with defects, while GLP-1 treatment (which has no impact on energy balance) increases the percentage of normal sperm. If a link exists between the modification of the expression of ORGs and the sperm quality, I would have expected comparable results for CR and GLP-1. How do the authors explain this? This point is never addressed in the manuscript.

Answer 3: We acknowledge the reviewer's comment on this subject. We decided to dedicate this work to the study of the impact of CR and GLP-1 administration on the metabolism, ORG expression, and fertility of animals since these two strategies could be used to improve metabolic health. We reported that CR had an impact on energy balance, while GLP-1 administration did not. The metabolic alteration imposed by CR on metabolism resulted in an increased percentage of sperm head defects. This is expected since the reproductive system is influenced by metabolic alterations. Meanwhile, no metabolic alterations were found in animals treated with GLP-1, which is reflected in an increased percentage of normal morphology sperm. These results indicate that GLP-1 therapy has the potential to improve fertility parameters, without alterations in the energy balance. Moreover, due to these initial results, we would not expect that the expression of ORG in the testes and sperm from CR and GLP-1 animals would be similar. We observed an overall increase in the ORG expression in the testis of animals from the two groups. Regardless, the expression of these genes reveals to have very different profiles between treatments. We could only find a positive correlation between the abundance of MC4R transcript in spermatozoa from animals treated with GLP-1 and the abundance of NFE2L2 mRNA, suggesting that MC4R could, somehow, contribute the oxidative stress regulation. Meanwhile, the expression of ORG in testes and sperm from animals exposed to CR was correlated to several other parameters. These results were to be expected, taking into consideration that CR induced more significant metabolic alterations than the GLP-1 administration.

Issue 4: The authors also speculate on a possible modulation of oxidative stress following the treatment, but the total antioxidant capacity is not modified. The highlighted increase in nfr2 expression remains a very indirect clue. As before, if the effect on oxidative capacity is comparable for CR and GLP-1 (increased expression of nfr2), how can the authors reconcile this result with the differential effects of the 2 treatments on sperm quality. Based on the results, the following sentence is incorrect and should be rewritten. "Overall, we were able to unveil that CR is associated with increased ORGs expression and improvement in oxidative status both in tests and sperm." CR does not modify nfr2 in sperm. Regarding the effect of GLP-1, how do the authors envisage its action in the testicles? Is this a direct effect? Are there GLP-1 receptors at this level? The authors should discuss this.

Answer 4: We thank the reviewer for highlighting this subject. To address the issue raised, we begin by rewriting the incorrect sentence in question to " Overall, our results suggest an association between CR, the increase of ORGs expression and improvement of in oxidative status both in testes and sperm". In regards to the expression of the GLP-1R receptor, its presence was already identified in both the testes and spermatozoa. It is difficult to determine the effect that GLP-1 has on the testes. Previous works have reported that the exposure of Sertoli cells to GLP-1 decreased glucose consumption while increasing lactate production. Meanwhile, no effects were detected in mitochondria functionality in human SCs exposed to the GLP-1 levels found in healthy individuals. These results could indicate a beneficial effect of GLP-1 on the testes' metabolism.

Reviewer 2 Report

This is a complete study about the diet treatments against Obesity-related gene expression in sperm and testis. Overall its a good work, however, the authors need to address below comments before further decision.

General comments:

So many grammatical mistakes, please revise.

Specific comments:

Calorie restriction (CR) diets and GLP-1 analogue drugs and are: Check grammar

 GLP-1 analogue: Expand

 CR and GLP-1 administration is: Check grammar

 CR (n=6): Dose

Controls (n=5): Explain the treatment

ORGs expression in spermatozoa and testis: Breifly explain why did you expect to have relation between ORG and male fertility, why you are specifically looking for ORGs expression in spermatozoa and testis not in fat tissues? 

 fat mass and obesity-associated (FTO): why abbreviated as FTO?

was evaluated in testis and spermatozoa.: Explain the techniques used?

FTO and TMEM18 expression in testis and MC4R and TMEM in sperm were:Reframe

Sperm FTO and TMEM expression were:Reframe

FTO and TMEM18 expression in testis: Explain the treatment CR or GLP-1 analogue drugs treated?

Sperm FTO and TMEM18 expression were positively: Same, treatment details?

 CR and GLP1 administration results in higher ORGs expression: To claim this, clearly state your results data.

Assessment of sperm-quality parameters: It is better to assess sperm motility against different treatment.

 from the rats’ testes and sperm the NYZ Total RNA Isolation: Incomplete

Table 1.: Wrong format, remove vertical line in table and add borders on top and bottom of table. Refer standard format.

Table 2: Upper border is missing

Table 3, 4.: Check format

Figure 2: Poor resolution

Figure 3: Scale bar

Figures numbers were misordered throughout, carefully check figure captions and order. For ex.

Figure 1: Effect GLP-1 administration: This is not Fig.1

Figure 2: Expression of NFE2L2: Provide high resolution gel image

 CR and GLP-1 administration could impact on male reproductive function by altering sperm quality: Try to explain the mechanism behind this effect. How did the CR and GLP-1 administration affect responsible genes expression?

In general, very long discussion, cut out general stories and concise the part.

Also try to shorten Conclusions.

Author Response

First, we would like to acknowledge the Reviewers for the dedicated analysis of the manuscript and the questions addressed to the different subjects, in order to improve the understanding of experimental work and the rationalization of its results. Thereby, we dissected the questions and, point-by-point, we answered them, signalizing in yellow the sentences or sections that were altered in the manuscript.

Reviewer #2

General comment: This is a complete study about the diet treatments against Obesity-related gene expression in sperm and testis. Overall it’s a good work, however, the authors need to address below comments before further decision.

Answer: We thank the reviewer for these encouraging words. We thank the reviewer for all comments that helped to improve our manuscript.

Issue 1: So many grammatical mistakes, please revise.

Answer 1: Thank you for your comment. I have revised the language of the entire manuscript. We hope we have elevated the level of our writing.

Issue 2: Calorie restriction (CR) diets and GLP-1 analog drugs and are: Check grammar

Answer 2: We have checked the language throughout the entire manuscript. Thank you.

Issue 3: GLP-1 analogue: Expand

Answer 3: Throughout this manuscript, we mention several times the words: GLP-1 analogs therapies. As is mentioned in the Introduction of our manuscript: “longer-acting GLP‐1 analogs have been developed to be used as pharmacological tools for type 2 diabetes (T2D) and obesity treatment [9]. In fact, the glucose-lowering and weight loss efficacy of GLP-1 analogs, such as liraglutide and semaglutide, is well demonstrated, besides improving glucose tolerance and decreasing insulin resistance [10-12]”. In resume, several authors are studying the effects of GLP-1 through these GLP-1 analogs. As mentioned in our manuscript, GLP-1 has a very short half-life time. Because of this, the study of these analogs is often easier and provides information for future therapies to treat metabolic disorders. Thank you for your comment.

Issue 4: CR and GLP-1 administration is: Check grammar

Answer 4: We have checked the language throughout the entire manuscript. Thank you.

Issue 5: CR (n=6): Dose

Answer 5: CR stands for caloric restriction. The animals exposed to caloric restriction received 30% less chow diet than the former for 28 days in comparison to the control group, as explained in the methods section of our manuscript. Thank you for your comment.

Issue 6: Controls (n=5): Explain the treatment

Answer 6: It is described in the methods section of our manuscript: “5 (animals) were used as controls (CTR); 5 were subjected to peritoneal implantation of a mini‐pump for GLP‐1 delivery at a constant rate of 3.5 pmol/min/kg for 28 days, and 6 were subjected to caloric restriction (CR). Rats from the CTR and GLP-1 groups were fed ad libitum with a standard chow diet (4RF21 certificate, Mucedola, Italy), whereas the rats in the CR group received 30% less chow diet than the former for 28 days”. Thank you for your comment.

Issue 7: ORGs expression in spermatozoa and testis: Briefly explain why did you expect to have relation between ORG and male fertility, why you are specifically looking for ORGs expression in spermatozoa and testis not in fat tissues?

Answer 7: We aimed to investigate the potential impact of negative energy balance promoted by CR and GLP-1 administration on male reproductive function, and we hypothesized that the four studied ORGs might be mediating some of the CR and GLP-1 administration effects in testes or sperm, that influence the male fertility. This is because, as we mentioned in the manuscript, it has already been described that the expression of these ORGs responds to hormonal imbalance (namely the one promoted by CR and GLP-1 administration). Their abundance has also been associated with sperm quality and consequently with male fertility. In addition, since this topic is little or even not described in the literature, we wanted to contribute to this knowledge.

Issue 8: fat mass and obesity-associated (FTO): why abbreviated as FTO?

Answer 8: We abbreviate the Fat mass and obesity gene as FTO since this gene is presented this way in the literature, although the letters of the acronym do not effectively correspond to the gene name. According to Fawcett and Barroso (2010), the FTO gene was the first locus unequivocally associated with adiposity, even though the function of the gene was known at the time (2007). This is the reason, most like, for this locus to be known as Fat mass and obesity gene, and not as the protein it encodes, an alpha-ketoglutarate-dependent dioxygenase enzyme. We hope that this information answers the reviewer's question. Thank you.

Issue 9: was evaluated in testis and spermatozoa.: Explain the techniques used?

Answer 9: The techniques used were added to the abstract and are described in detail in the methods section of our manuscript. Thank you for your comment.

Issue 10: FTO and TMEM18 expression in testis and MC4R and TMEM in sperm were: Reframe

Answer 10: The sentence in question was altered. Thank you for the suggestion.

Issue 11: Sperm FTO and TMEM expression were: Reframe

Answer 11: The sentence in question was altered. Thank you for the suggestion.

Issue 12: FTO and TMEM18 expression in testis: Explain the treatment of CR or GLP-1 analogue drugs treated?

Answer 12: The sentence in question was altered, and the information requested was added. Thank you for the suggestion.

Issue 13: Sperm FTO and TMEM18 expression were positively: Same, treatment details?

Answer 13: The sentence in question was altered, and the information requested was added. Thank you for the suggestion.

Issue 14: CR and GLP1 administration results in higher ORGs expression: To claim this, clearly state your results data.

Answer 14: The sentence in question was already altered, according to what was suggested by reviewer 1. It is now written: “CR and GLP-1 administration results in higher ORGs expression in testes, and these were correlated to several alterations on sperm fertility parameters”. We believe that the new conclusion better suits our results. Thank you for the suggestion.

Issue 15: Assessment of sperm-quality parameters: It is better to assess sperm motility against different treatment.

Answer 15: Sperm motility was not assessed in this work, since the animals were deeply anesthetized with carbon dioxide before sacrifice. Since we are dealing with large animals, anesthesia is a necessary step to preserve the well-being of the animals. However, it has a drastic effect on the motility of spermatozoa. Due to this motive, motility was not assessed. Thank you for your comment.

Issue 16: from the rats’ testes and sperm the NYZ Total RNA Isolation: Incomplete

Answer 16: The isolation of RNA was performed according to the NYZ Total RNA Isolation manufacturer’s (NZYTech, Lisbon, Portugal) instructions. Thank you for your comment.

Issue 17: Table 1.: Wrong format, remove vertical line in table and add borders on top and bottom of table. Refer standard format. Table 2: Upper border is missing. Table 3, 4.: Check format. Figure 2: Poor resolution. Figure 3: Scale bar. Figures numbers were misordered throughout, carefully check figure captions and order. For ex. Figure 1: Effect GLP-1 administration: This is not Fig.1

Answer 17: All the issues were addressed. Thank you for all the suggestions.

Issue 18: Figure 2: Expression of NFE2L2: Provide high resolution gel image

Answer 18: The original gel images will be added to the supplemental data of this manuscript. Thank you for your suggestion.

Issue 19: CR and GLP-1 administration could impact on male reproductive function by altering sperm quality: Try to explain the mechanism behind this effect. How did the CR and GLP-1 administration affect responsible genes expression?

Answer 19: Genetic variations related to energy metabolism, physical activity, appetite control, and the utilization of dietary components play an important role in the response to nutritional interventions. In addition to the significant associations between specific polymorphisms and outcomes of CR – nutrigenetics - there is increasing evidence that energy intake influences gene (mRNA levels) expression – nutrigenomics. In fact, CR has been shown to affect longevity and metabolism, and its hypothesize that these changes are mediated in part by a differential gene expression, namely those associated with energy homeostasis. The normal male reproductive function depends on an adequate nutritional state, namely for processes such as spermatogenesis. Therefore, it is expectable that a negative energy balance, such as promoted by CR and GLP-1 administration might lead to effects on male fertility, namely by altering parameters related to sperm quality. In fact, one of our authors described that metabolic dynamics of human Sertoli cells are differentially modulated by physiological and pharmacological concentrations of GLP-1, being that GLP-1 receptors are expressed in Sertoli cells.

Marti, A., Martinez-Gonzalez, M. A. & Martinez, J. A. Interaction between genes and lifestyle factors on obesity. Proceedings of the Nutrition Society 67, 1-8 (2008).

Redman, L. M., Martin, C. K., Williamson, D. A. & Ravussin, E. Effect of caloric restriction in non obese humans on physiological, psychological and behavioral outcomes. Physiology & Behavior 94, 643-648 (2008).

Issue 20: In general, very long discussion, cut out general stories and concise the part. Also try to shorten Conclusions.

Answer 20: We have shortened both sections as much as we could, without deleting any information that we considered important. Thank you for the suggestion.

Round 2

Reviewer 1 Report

I appreciate the authors for their responses and revisions to the manuscript.

Reviewer 2 Report

The revised version is satisfactory.